# Monitoring of Dabrafenib and Trametinib in Serum and Self-Sampled Capillary Blood in Patients with BRAFV600-Mutant Melanoma

**DOI:** 10.3390/cancers14194566

**Published:** 2022-09-20

**Authors:** Nora Isberner, Anja Gesierich, David Balakirouchenane, Bastian Schilling, Fatemeh Aghai-Trommeschlaeger, Sebastian Zimmermann, Max Kurlbaum, Alicja Puszkiel, Benoit Blanchet, Hartwig Klinker, Oliver Scherf-Clavel

**Affiliations:** 1Department of Internal Medicine II, University Hospital Würzburg, Oberdürrbacher Strasse 6, 97080 Würzburg, Germany; 2Department of Dermatology, Venerology and Allergology, University Hospital Würzburg, Josef-Schneider-Strasse 2, 97080 Würzburg, Germany; 3Department of Pharmacokinetics and Pharmacochemistry, Cochin Hospital, AP-HP, Cancer Research for Personalized Medicine (CARPEM), 75014 Paris, France; 4Faculty of Pharmacy, Paris Cité University, CiTCoM, 8038 CNRS, Inserm U1268, 75006 Paris, France; 5Institute of Pharmacy and Food Chemistry, University of Würzburg, Am Hubland, 97074 Würzburg, Germany; 6Department of Internal Medicine I, University Hospital Würzburg, Oberdürrbacher Strasse 6, 97080 Würzburg, Germany; 7Core Unit Clinical Mass Spectrometry, University Hospital Würzburg, Oberdürrbacher Strasse 6, 97080 Würzburg, Germany; 8Faculty of Pharmacy, Paris Cité University, Inserm UMR-S1144, 75006 Paris, France

**Keywords:** dabrafenib, trametinib, hydroxy-dabrafenib, melanoma, BRAF mutation, volumetric absorptive micro-sampling (VAMS), at-home sampling, drug monitoring, population pharmacokinetics

## Abstract

**Simple Summary:**

In melanoma patients treated with dabrafenib and trametinib, dose reductions and treatment discontinuations related to adverse events (AE) occur frequently. However, the associations between patient characteristics, AE, and exposure are unclear. Our prospective study analyzed serum (hydroxy-)dabrafenib and trametinib exposure and investigated its association with toxicity and patient characteristics. Additionally, the feasibility of at-home sampling of capillary blood was assessed, and a model to convert capillary blood concentrations to serum concentrations was developed. (Hydroxy-)dabrafenib or trametinib exposure was not associated with age, sex, body mass index, or AE. Co-medication with P-glycoprotein inducers was associated with lower trough concentrations of trametinib but not (hydroxy-)dabrafenib. The applicability of the self-sampling of capillary blood was demonstrated. Our conversion model was adequate for estimating serum exposure from micro-samples. The monitoring of dabrafenib and trametinib may be useful for dose modification and can be optimized by at-home sampling and our new conversion model.

**Abstract:**

Patients treated with dabrafenib and trametinib for BRAF^V600^-mutant melanoma often experience dose reductions and treatment discontinuations. Current knowledge about the associations between patient characteristics, adverse events (AE), and exposure is inconclusive. Our study included 27 patients (including 18 patients for micro-sampling). Dabrafenib and trametinib exposure was prospectively analyzed, and the relevant patient characteristics and AE were reported. Their association with the observed concentrations and Bayesian estimates of the pharmacokinetic (PK) parameters of (hydroxy-)dabrafenib and trametinib were investigated. Further, the feasibility of at-home sampling of capillary blood was assessed. A population pharmacokinetic (popPK) model-informed conversion model was developed to derive serum PK parameters from self-sampled capillary blood. Results showed that (hydroxy-)dabrafenib or trametinib exposure was not associated with age, sex, body mass index, or toxicity. Co-medication with P-glycoprotein inducers was associated with significantly lower trough concentrations of trametinib (*p* = 0.027) but not (hydroxy-)dabrafenib. Self-sampling of capillary blood was feasible for use in routine care. Our conversion model was adequate for estimating serum PK parameters from micro-samples. Findings do not support a general recommendation for monitoring dabrafenib and trametinib but suggest that monitoring can facilitate making decisions about dosage adjustments. To this end, micro-sampling and the newly developed conversion model may be useful for estimating precise PK parameters.

## 1. Introduction

Molecularly targeted therapy is part of the standard of care in patients with advanced and metastatic BRAF^V600^-mutant melanoma worldwide [1]. The commonly used combination of dabrafenib and trametinib acts by dual inhibition of the mitogen-activated protein kinase (MAPK) pathway suppressing BRAF and MEK, respectively [2,3]. Dual therapy with dabrafenib and trametinib has proven to prolong progression-free and overall survival compared to BRAF inhibiting monotherapy with dabrafenib or vemurafenib [4,5].

However, treatment efficacy varies greatly across patients. For example, in a phase 3 trial, treatment was permanently discontinued for 13% of patients, and dosages were reduced due to adverse events (AE) for 33% [4]. Dabrafenib and trametinib are administered at a standard fixed dose despite high interindividual pharmacokinetic variability and existing evidence for the benefits of therapeutic drug monitoring (TDM) and precision dosing of oral targeted cancer therapies [6,7,8]. Relevant interindividual variability in drug exposure has particularly been described for dabrafenib [9,10,11], which undergoes metabolism via the cytochrome P450 (CYP) enzymes CYP3A4 and CYP2C8 to form the active metabolites hydroxy-, carboxy-, and desmethyl-dabrafenib. Only hydroxy-dabrafenib has been considered relevant for the pharmacodynamic activity of dabrafenib [2,12,13]. Genetic polymorphisms and co-medication with inhibiting or inducing drugs are well-described causes of altered CYP expression [14]. Lower pharmacokinetic (PK) variability has been observed for trametinib as metabolism is mediated by hydrolytic enzymes less prone to interindividual variability [3,9,10,12,15]. Both compounds are substrates of the multidrug transporter P-glycoprotein (P-gp) [2,3,16,17]. Additionally, the administration of both compounds with food results in decreased bioavailability [2,3].

Present knowledge of the associations of age, sex, and weight with exposure remains contradictory [5,12,15,18,19]. Likewise, investigations on exposure–efficacy relationships have revealed inconsistent results regarding the associations between progression-free survival, duration of response, and overall survival and dabrafenib or trametinib exposure [9,10,12,15,19,20,21,22]. The same holds true for evidence on the associations of exposure with toxicity. While some studies suggest that AE occur more frequently in patients with elevated dabrafenib [9,12,23] or trametinib [10] exposure, an exposure–toxicity relationship could not be confirmed by other investigations [9,10,12,19,24].

However, some investigations used observed trough concentrations (C_min_) to analyze the effects of covariates and exposure-toxicity relationships [9,19,23]. In contrast, others used the population pharmacokinetic (popPK) model to predict the area under the curve (AUC) or both [10,12,24]. A previously published popPK model for dabrafenib has shown that C_min_ is not dose-related [18], suggesting that AUC may be the more precise parameter to investigate exposure. In routine outpatient care, collecting trough samples using venipuncture is the most common procedure. PopPK models are helpful tools to generate empirical Bayesian estimates (EBE) for AUC and other PK parameters based on trough or untimed samples [25,26,27,28]. However, if the model is only informed by a few trough concentrations or samples at similar time intervals, the results might lack accuracy [29,30]. Micro-sampling techniques have the potential to overcome this obstacle, as the sampling of capillary blood can be performed repeatedly at different sampling times by the patient at home [31,32,33,34]. To the best of our knowledge, no studies on monitoring of dabrafenib and trametinib using capillary blood samples have been published so far.

We conducted a prospective real-life study to assess the pharmacokinetics of dabrafenib, hydroxy-dabrafenib, and trametinib and to characterize the exposure–toxicity relationships in patients with advanced and metastatic BRAF^V600^-mutant melanoma. The feasibility of micro-sampling for use in clinical routine was evaluated using the volumetric absorptive micro-sampling (VAMS) device. Serum and capillary blood samples were used to develop a popPK model-informed VAMS-to-serum conversion model to describe individual PK parameters based on self-sampled capillary blood concentrations.

## 2. Materials and Methods

### 2.1. Patient Selection

Patients treated with dabrafenib and/or trametinib for advanced or metastatic BRAF^V600^-mutant melanoma (AJCC stage III and IV) at the Department of Dermatology, Venerology and Allergology of the University Hospital of Würzburg were enrolled in this prospective observational study between July 2019 and May 2021. Patients were included independently from the dosing regimen or duration of treatment. Exclusion criteria were age < 18 years and estimated life expectancy < 2 months, according to the treating physician. The study was approved by the Ethics Commission of the University of Würzburg (ref 199/18-am). All performed procedures were in accordance with the declaration of Helsinki. Written informed consent was obtained from all patients. 

### 2.2. Collection of Serum Samples

Serum samples were collected regularly during each routine outpatient visit from study enrollment until January 2022 or until treatment discontinuation. If possible, blood sampling was performed before the next drug administration, but any time interval between the last administration and blood withdrawal was accepted. The steady-state was assumed after continuous dosing for fourteen days for both compounds. Fourteen days was chosen for dabrafenib despite its short half-life of eight hours due to known CYP3A autoinduction with an induction half-life of approximately 67 h [2,18].

### 2.3. Collection of Capillary Blood Samples

In addition to serum sampling, patients were asked to participate in the sampling of capillary blood in the clinic and at home. Micro-sampling was not obligatory for enrollment in the study. Micro-sampling was performed from a finger by drawing 20 µL of capillary blood onto the VAMS device MITRA^®^ purchased from Neoteryx (Torrance, CA, USA). During outpatient visits, micro-sampling was conducted by medical staff in parallel to serum sampling. A maximum of six corresponding serum/VAMS samples were collected per patient. Additionally, patients were asked to perform at-home sampling after being instructed by trained staff. For each at-home sampling occasion, patients received a set including four VAMS devices. Patients were instructed to perform sampling at four different time points during the following four time slots: 9 a.m.–12 p.m., 12–3 p.m., 3–6 p.m., and 6–9 p.m. Time slots instead of defined time points were given as performing sampling at predetermined times of day has proven to be inconvenient for the implementation of micro-sampling in a patient’s daily routine. Samples collected during different time slots were still accepted. The time of last drug administration and sampling times were self-documented on a specific form. No special storage conditions were specified, and patients were asked to keep samples at ambient temperature until shipment. Samples were sent to the laboratory in an airtight and opaque bag containing desiccant for analysis. At-home samples were collected at up to four occasions per patient. Hematocrit was documented from the results of routine blood withdrawal for VAMS samples collected during outpatient visits. For at-home samples, the last known hematocrit was used.

### 2.4. Assessment of Adverse Events

AE was routinely assessed by the treating physicians at each serum sampling time point and graded according to the Clinical Terminology for Adverse Events Criteria (CTCAE) Version 5.0 [35]. Additionally, permanent and temporary treatment discontinuations, dose reductions, and corresponding reasons were documented. Further recorded patient data included age, sex, height, weight, smoking status, dosing regimen, co-medication, and relevant laboratory parameters if ordered by the treating physician (platelet count, white blood cell count, absolute neutrophil count, absolute lymphocyte count, hemoglobin, bilirubin, aspartate and alanine aminotransferase, alkaline phosphatase, gamma-glutamyl transferase, amylase, lipase, creatine kinase, phosphate, sodium, glucose, serum creatinine, and estimated glomerular filtration rate). The treatment and frequency of visits were managed by the treating physicians without knowing the measured drug concentrations. AE was not assessed simultaneously with the collection of VAMS samples.

### 2.5. Quantification of Dabrafenib, Hydroxy-Dabrafenib, and Trametinib in Serum and Capillary Blood

Dabrafenib and trametinib serum concentrations were quantified using a fully validated liquid chromatography-tandem mass spectrometry method (LC-MS/MS) [36]. The lower level of quantification (LLOQ) of the method was 6 and 2 ng/mL for dabrafenib and trametinib, respectively. Hydroxy-dabrafenib concentrations were analyzed using a different previously published LC-MS/MS method with an LLOQ of 10 ng/mL [37]. A highly specific and fully validated LC-MS/MS assay was used for the analysis of the capillary blood concentrations of dabrafenib and trametinib, with the LLOQ being 6 and 2 ng/mL, respectively [38]. Hydroxy-dabrafenib was not quantified in capillary blood samples as the methodology was not available at the time of analysis. All the methods have been validated according to the guidelines of the U.S. Food and Drug Administration and/or the European Medicines Agency.

### 2.6. Data Processing and Statistical Analysis

Data were collected and processed using Microsoft Excel 2016 Version 16.0 (Microsoft Corporation, Redmond, WA, USA). Statistical calculations and visualization of results were performed with R Studio Version 1.2.5042 (RStudio Incorporation, Boston, MA, USA) running R version 4.0.5 (R Foundation for Statistical Computing, Vienna, Austria, 2020). For further analysis, serum concentrations were divided into groups according to time interval post-dose as follows: 0–5 h, 5–10 h, 10–14 h, and >14 h for (hydroxy-)dabrafenib and 0–10 h, 10–20 h, 20–30 h, and >30 h for trametinib. Concentrations < LLOQ were excluded from further statistical analyses. The concentrations of the same patient at the same dosage and within the same post-dose group were summarized into an individual mean serum concentration for descriptive statistical analysis to account for potential bias caused by the unbalanced number of samples per patient. Concentrations stratified by dosing regimen were also analyzed across all patients to demonstrate the entire degree of variability. The Wilcoxon signed-rank test was used for unpaired samples. The correlations between individual PK parameters sampled from the conditional distribution during the generation of EBE (methods for estimation see below) and age, sex, body mass index (BMI), and co-medication were analyzed using analysis of variance (ANOVA) and Spearman’s correlation coefficient. Logistic regression was used to compare the risk of treatment discontinuations or dose reductions (dabrafenib dose reduction or discontinuation, trametinib dose reduction or discontinuation, or any dose reduction or discontinuation) with the corresponding predicted trough concentrations and area under the curve of a dosing interval (AUC_τ_) of dabrafenib, hydroxy-dabrafenib, and trametinib as well as with the composite predicted dabrafenib/hydroxy-dabrafenib trough concentration and with composite dabrafenib/hydroxy-dabrafenib AUC_τ_ assuming standard dosing. The Hosmer–Lemeshow test was used to evaluate the goodness of fit for our logistic regression model. Due to the explorative nature of the study, *p*-values obtained from multiple comparisons were not corrected for multiple testing. A *p*-value < 0.05 was considered statistically significant.

### 2.7. Estimation of Pharmacokinetic Parameters Based on Existing Population Pharmacokinetic Models

For each compound, a popPK model developed by Balakirouchenane et al. was assessed regarding its suitability to create EBE using Monolix 2021R1 (Lixoft SAS, Antony, Fance) [12]. Other models were not tested since they did not include hydroxy-dabrafenib and were not based on data derived from a real-world cohort [10,11]. The predictive performance of the models was estimated by calculating the prediction error (*PE*), mean prediction error (*MPE*), mean absolute prediction error (*MAPE*), and mean relative deviation (*MRD*):(1)$$PE_{i}\left[\%\right]=\frac{C_{predicted,i}-C_{observed,i}}{C_{observed,i}}\cdot 100$$
(2)$$MPE\left[\%\right]=\frac{1}{n}\overset{n}{\underset{i=1}{\sum }}PE_{i}$$
(3)$$MAPE\left[\%\right]=\frac{1}{n}\overset{n}{\underset{i=1}{\sum }}\left|PE_{i}\right|$$
(4)$$MRD=10^{x};x=\sqrt{\frac{1}{n}\overset{n}{\underset{i=1}{\sum }}{\left(log_{10}\left[C_{predicted,i}\right]-log_{10}\left[C_{observed,i}\right]\right)}^{2}}$$

The abbreviations in Equations (1)–(4) are as follows: *C_predicted,i_* = predicted serum concentration, *C_observed,i_* = corresponding observed serum concentration, *n* = number of observed values.

Estimates for AUC_τ_ were generated for every occasion for the standard doses (150 mg q12h for dabrafenib, 2 mg q24h for trametinib). In addition, the individual predicted C_min_ at a steady state was generated since not all patients contributed trough samples. The same estimates were used in the logistic regression model.

### 2.8. Development of a New Population Pharmacokinetic Model-Informed VAMS-to-Serum Conversion Model

The popPK models published by Balakirouchenane et al. [12] were extended to predict dabrafenib and trametinib serum concentrations from VAMS concentrations. The conversion was based on a hematocrit-dependent formula (Figure 1). Since the protein binding of dabrafenib is >99.5%, the impact of partitioning into blood cells can be neglected in the simplified conversion formula, according to Iacuzzi et al. [39]. Thus, erythrocytes only dilute the sample and do not contain relevant amounts of dabrafenib. For trametinib, VAMS concentrations were substantially higher compared to plasma concentrations, indicating the sequestration of the analyte into or onto the surface of blood cells. As proposed by Iacuzzi et al. [39], the blood cell to plasma partition coefficient (K_bp_) and intraindividual variability (IIV) were included in the conversion model and estimated from the consolidated data (Figure 1). For this purpose and to evaluate the conversion models, all the serum (Appendix A) and VAMS concentrations (collected at home or in the clinic) (Appendix A) were consolidated in a single dataset and fitted to the combined model using the EBE in Monolix 2021R1 (Lixoft SAS, Antony, France). Incorrectly sampled VAMS were removed from the dataset. The resulting individual predicted VAMS concentrations were compared with the observed VAMS concentrations. In the second step, the models were used to predict EBE for serum PK parameters solely from the at-home collected VAMS samples to demonstrate the feasibility of at-home sampling for deriving PK parameters. For trametinib, at least one paired sample per patient was included to generate the individual estimate for K_bp_. A visual predictive check was performed to evaluate the predictive performance of the method.

## 3. Results

### 3.1. Patient and Sample Characteristics

First, 27 patients were included, comprising 12 patients receiving dabrafenib and trametinib for the treatment of AJCC stage III and 15 patients receiving dabrafenib and trametinib for the treatment of AJCC stage IV BRAF^V600^-mutant melanoma. The median duration of treatment at enrollment was 146 days (range 11–1494 days), and the median inclusion time in our study was 324 days (range 26–714 days). No patient had signs of moderate or severe renal or hepatic impairment at the time of sampling. Further details on patient demography are presented in Table 1. In total, 278 serum samples were analyzed for (hydroxy-)dabrafenib and 266 serum samples for trametinib. The median number of samples per patient was 10 (IQR 8, range 1–22) and 8 (IQR 8.5, range 1–22), respectively. A total of 270 of the 278 (hydroxy-)dabrafenib serum samples were obtained at the standard daily dose of 150 mg q12h, and 214 of the 266 trametinib samples were obtained at the standard dose of 2 mg q24h. Two serum samples were excluded due to being below the LLOQ: one trametinib sample collected 30 min post-dose at 2mg, and one dabrafenib sample collected 12 h post-dose at 75 mg. Additionally, 18 patients contributed VAMS samples. Taking the VAMS samples collected in the clinic and at home together, 169 VAMS concentrations for dabrafenib and 158 VAMS concentrations for trametinib were analyzed. Among the VAMS devices, 95.3% were sampled correctly. In total, eight VAMS samples were removed from our dataset because the VAMS devices were not completely soaked with blood. All eight incorrectly collected samples were collected at home. More information on the sample characteristics and dosing regimens are presented in Appendix A.

### 3.2. Observed Dabrafenib, Hydroxy-Dabrafenib, and Trametinib Serum Concentrations

The observed mean dabrafenib, hydroxy-dabrafenib, and trametinib concentrations per patient stratified by time interval are presented in Figure 2 and Figure 3 and Appendix A. For patients on a dabrafenib dose of 150 mg q12h, the median of individual mean trough concentration was 45.0 ng/mL (IQR: 25.0 ng/mL, range: 20–155 ng/mL) for dabrafenib and 76.0 ng/mL (IQR: 51.4 ng/mL, range: 37.2–171.0 ng/mL) for hydroxy-dabrafenib. For trametinib, the median of mean trough concentration was 11.2 ng/mL (IQR: 2.40 ng/mL, range: 6.22–15.9 ng/mL) for 2 mg q24h.

For patients taking the standard dose, no significant differences in dabrafenib, hydroxy-dabrafenib, or trametinib steady-state serum trough concentrations were observed in terms of sex (*p* = 0.174 for dabrafenib, *p* = 0.837 for hydroxy-dabrafenib, *p* = 0.692 for trametinib), age (above and below 65 years, *p* = 1.0 for dabrafenib, *p* = 0.252 for hydroxy-dabrafenib, *p* = 0.865 for trametinib) and BMI (above and below 30 kg/m^2^, *p* = 0.377 for dabrafenib, *p* = 0.583 for hydroxy-dabrafenib, *p* = 0.219 for trametinib). Further details on the associations between observed concentrations and covariates are presented in Appendix A. Co-medication (Appendix A) with at least one moderate CYP2C8 inhibitor despite trametinib neither resulted in increased individual mean trough concentrations of dabrafenib (*p* = 0.689) nor of hydroxy-dabrafenib (*p* = 0.864). Trametinib is a strong CYP2C8 inhibitor but was not included, as it was co-administered in 95.3% of sampling occasions. Co-medication with CYP3A4 perpetrators was not tested due to the low number of samples derived from patients receiving medication interacting with CYP3A4 (one strong and one moderate CYP3A4 inhibitor in total). Concomitant administration of at least one P-gp inhibitor did not lead to significant differences in individual mean trough concentrations of dabrafenib (*p* = 0.351), hydroxy-dabrafenib (*p* = 0.351), or trametinib (*p* = 0.657), whereas comedication with at least one P-gp inducer led to a significant increase in trametinib individual mean trough concentrations (*p* = 0.027) but not of dabrafenib or hydroxy-dabrafenib concentrations (*p* = 0.594 and *p* = 0.099, respectively). No significant differences in dabrafenib, hydroxy-dabrafenib, or trametinib individual mean trough concentrations were observed in patients with or without proton pump inhibitors (*p* = 0.305 for dabrafenib, *p* = 0.171 for hydroxy-dabrafenib, *p* = 0.574 for trametinib). 

### 3.3. Estimates for Pharmacokinetic Parameters Based on Existing Population Pharmacokinetic Models

Generated population predictions for observed concentrations using patient covariates required in the respective model and dosing regimens are presented in Appendix A. Observed concentrations in our population and their variability were well captured by both models (Appendix A). Predictive performance was acceptable (Appendix A). The summarized estimates for AUC_τ_ and C_min_ at steady state (average over all observations per patient) are presented in Appendix A. For dabrafenib and hydroxy-dabrafenib, the median of the average predicted AUC_τ_ was 5944 ng. h/mL (IQR: 1389 ng. h/mL) and 6040 ng h/mL (IQR: 2220 ng. h/mL), respectively. The corresponding median of the average predicted trough concentrations was 42.3 ng/mL (IQR: 31.5 ng/mL) and 84.5 ng/mL (IQR: 68.0 ng/mL) for dabrafenib and hydroxy-dabrafenib, respectively. For trametinib, the median of the average predicted AUC_τ_ was 331.4 ng h/mL (IQR: 42.7), and the median of the average predicted trough concentrations were 11.5 ng/mL (IQR: 1.66 ng/mL). 

No significant correlation between sex, age (above and below 65 years), BMI (above and below 30 kg/m^2^), or co-medication and estimated PK parameters sampled from the conditional distribution was found.

### 3.4. Adverse Events, Dose Reductions and Treatment Discontinuations

All patients reported AE or displayed a potentially treatment-related laboratory abnormality at least once during the observation period. In total, 1198 adverse events (301 clinical adverse events and 897 laboratory abnormalities) were documented during 278 visits. Additionally, 2.2% of all documented AE were CTCAE grade 3. No AE of grade 4 or 5 occurred during the observation period. The clinical adverse events and laboratory abnormalities are summarized in Appendix A. AE leading to dose reductions during the study period was displayed by 11.1% of patients: Two patients (7.4%) had dabrafenib dose reductions (increase in aspartate and alanine aminotransferase, gamma glutamyl transferase, alkaline phosphatase, and creatine phosphokinase CTCAE grade 2; nausea and fatigue CTCAE grade 1) and one patient (3.7%) had a trametinib dose reduction (increase in creatine phosphokinase CTCAE grade 3). Additionally, when including dose reduction before study inclusion, 29.6% of patients had a dose reduction of dabrafenib and/or trametinib. In addition, 11.1% of patients required permanent treatment discontinuations; two patients (7.4%) discontinued dabrafenib and trametinib permanently (increase in lipase and gamma glutamyl transferase CTCAE grade 3 and of alkaline phosphatase CTCAE grade 2; pyrexia CTCAE grade 3) and one patient (3.7%) discontinued only trametinib permanently (reduced left ventricular ejection fraction and peripheral edema CTCAE grade 1). No permanent treatment discontinuations occurred beforehand (Appendix A).

Logistic regression revealed no significant relationships between dose reduction or discontinuation due to AE and predicted dabrafenib, hydroxy-dabrafenib, or trametinib serum trough concentrations or AUC_τ_ or predicted composite dabrafenib/hydroxy-dabrafenib serum trough concentrations or AUC_τ_. The lowest *p*-values were obtained for trametinib dose reduction or discontinuation vs. predicted trametinib trough concentration (*p* = 0.08) and predicted trametinib AUC_τ_ (*p* = 0.10) (Appendix A). The Hosmer–Lemeshow test showed that our logistic regression model was suitable (trametinib dose reduction or discontinuation vs. predicted trametinib trough concentration, *p* = 0.52; trametinib dose reduction or discontinuation vs. predicted trametinib AUC_τ_, *p* = 0.07).

### 3.5. Estimates for Serum Pharmacokinetic Parameters from Self-Sampled Capillary Blood

For the extended trametinib model, a population value for K_bp_ of 4.62 (RSE: 8.19%) with an IIV of ω_Kbp_ of 0.31 (RSE: 19.8%) was estimated. The newly developed popPK model-informed VAMS-to-serum conversion model was used to simulate VAMS concentrations for each occasion. Conversion models for both compounds resulted in an acceptable prediction of measured concentrations (Figure 4 and Appendix A). The derived PK profiles displayed very good reproducibility between occasions (Appendix A). Only for DT002 were the individually measured trametinib concentrations not captured well by the estimated concentration time curves (Appendix A). The EBE for serum PK parameters generated for each at-home sampling occasion for each patient is presented in Table 2 and Table 3.

For dabrafenib, an average AUC_τ_ was generated from different occasions since the model allowed inter-occasion variability, whereas trametinib estimates did not differ between occasions as inter-occasion variability was not included in the model. DT026 was the only patient with a reduced daily dabrafenib dose of 200 mg. The simulated AUC_τ_ of this patient at the standard dose of 300 mg daily was lower than the AUC_τ_ of patients receiving 300 mg daily (Table 2). Two of the seven trametinib patients contributed VAMS samples at a reduced daily dose of 1 mg (DT018) and 1.5 mg (DT019). The simulated AUC_τ_ of these patients at the standard daily dose of 2 mg was higher in comparison to the AUC_τ_ of patients who were stable on 2 mg q24h (Table 3).

## 4. Discussion

The observed dabrafenib, hydroxy-dabrafenib, and trametinib trough serum concentrations in our study population are in line with previously reported concentrations in real-life and clinical trial cohorts [9,10,15,19,22,23,24] and using a popPK model previously published by Balakirouchenane et al. [12] for generating EBE of AUC_τ_, and C_min_ resulted in an adequate description of dabrafenib, hydroxy-dabrafenib, and trametinib PK parameters in our cohort.

We did not observe significant correlations between observed serum trough concentrations or individual PK parameters and sex, age, or BMI. Furthermore, we did not find a relationship between the occurrence of dose reductions or treatment discontinuations and predicted exposure to dabrafenib, hydroxy-dabrafenib, and trametinib. Balakirouchenane et al. described [12] age as a relevant covariate for dabrafenib/hydroxy-dabrafenib composite exposure in their model, and Rousset et al. [9] found significantly higher trough concentrations in patients above 60 years of age. Ouellet et al. [18] did not include age in their dabrafenib model. While sex was described as a significant covariate for (hydroxy-)dabrafenib in Balakirouchenane et al.’s and Ouellet et al.’s models [12,18], Raynal et al. [19] did not observe significant differences in dabrafenib trough concentrations. For trametinib, Balakirouchenane et al.’s model did not find sex to be a relevant covariate, whereas Ouellet et al.’s model [15] found lower clearance in female patients and Raynal et al. observed higher trough concentrations in women. Only Ouellet et al.’s models [15,18] described weight as a relevant covariate for both compounds, but other models and real-life observations did not find an association [12,19]. Balakirouchenane et al. described a significantly higher AUC_τ_ of dabrafenib in patients requiring dose reductions due to toxicity. Rousset et al. found that trough concentrations of dabrafenib are significantly higher in patients with dose reductions and proposed a plasma trough threshold of 48 ng/mL. Goldwirt et al. and Raynal et al. found no correlation between AE and dabrafenib C_min_ or AUC_τ_ [10,19]. Additionally, Raynal et al. could not confirm the previously reported threshold of 48 ng/mL. One publication observed a nonsignificant trend between dabrafenib and hydroxy-dabrafenib exposure and pyrexia [24], whereas another study found no association [23]. For trametinib, only Goldwirt et al. [10] reported higher trametinib C_min_ and AUC_τ_ in patients experiencing any grade AE compared to patients without AE.

The comparability of the above-mentioned publications is limited by the differing study designs. The data for Ouellet’s models were derived from phase 1, 2, and 3 trials and might not reflect a typical real-life cohort. Some investigations used observed trough concentrations [9,19], whereas others included AUC_τ_ and other PK parameters [12,15,18]. Not all studies included relevant metabolites of dabrafenib in their investigations. Different approaches were used for analysis (e.g., age as a continuous covariate, patients above vs. below 60 years) to assess the effects of age on exposure. For investigation of exposure-toxicity relationships, some studies, including ours, used dose-limiting toxicity as an endpoint, whereas others included all AE. Our study design may have underestimated the association between exposure and toxicity because the median duration of therapy at enrollment was 146 days (range 11–1494 days), but many AE (e.g., pyrexia) occur close to therapy initiation [23]. This is supported by the number of toxicity-related dose reductions during the study period, which was less frequent in our population than in the cohort of the phase 3 trial (11.1 vs. 33%). However, the number of dose reductions in our cohort before and after study inclusion was like that of the phase 3 trial (29.6 vs. 33%) [4]. On the other hand, permanent treatment interruptions occurred only during the observation period and were almost as frequent in our cohort as in the phase 3 trial population (11.1% vs. 13%). Furthermore, the limited number of patients enrolled in our trial is a limitation to the logistic regression analysis. In addition, AE was only recorded during outpatient visits. Neither mild AE not requiring a patient visit to the clinic nor severe AE requiring hospital admission were recorded. Therefore, our data mostly reflect a population that is stable on therapy. Additionally, there was no documentation of patient-reported outcomes (PRO-CTCAE). The fact that neither age, sex, nor BMI was a significant covariate for the observed trough concentrations or model predicted AUC_τ_ or C_min_ of dabrafenib, hydroxy-dabrafenib, or trametinib in our real-life cohort suggests that one covariate alone is not decisive for dosage adjustments. Ouellet et al. also concluded that their observed effects of sex and weight on dabrafenib exposure are not clinically relevant [18]. However, most investigations regard each covariate independently. Future studies should investigate whether an unfavorable combination of several factors (e.g., low weight, high age, female) leads to relevant changes in exposure. Interestingly, Groenland et al. [20] proposed an efficacy target of 15.5 ng/mL for trametinib as patients with a C_min_ above 15.5 ng/mL showed significantly longer progression-free survival. Even though our observed concentrations were like those of previously published investigations, almost all our patients were below this target. Given that most of our patients are stable on therapy, our data suggest that most patients in our cohort would qualify for dose escalation to improve outcomes. Furthermore, the assessment of exposure–toxicity relationships is complicated by the use of combination therapy. All investigations have analyzed associations between AE and exposure for dabrafenib and trametinib separately. However, since dabrafenib and trametinib can cause similar AE, it is possible that low exposure to one substance might compensate for above-average exposure to the other substance or that exposure at the upper limit of normal to both compounds cause toxicity. Moreover, further investigations should be conducted in a larger cohort of patients to increase the power of statistical results. 

None of the other studies systematically investigated the role of co-medication in interacting with relevant enzymes and transporters. A substantial part of our study population received P-gp (53.6% received at least one inhibitor; 40.0% received at least one inducer) and CYP2C8 (35.6% received at least one inhibitor despite trametinib) perpetrators. In contrast, the effects of CYP3A4 could not be studied due to the low number of patients receiving CYP3A4 inhibiting or inducing co-medication. We were able to demonstrate a significant decrease in trametinib individual mean trough concentrations due to the concurrent administration of P-gp inducers but observed no effect on dabrafenib or hydroxy-dabrafenib. However, P-gp was not associated with estimated individual PK parameters. Co-medication with P-gp or CYP2C8 inhibitors did not change the exposure of any compound. The fact that our population mostly reflects patients that are stable on therapy might contribute to the underestimation of the co-medication effects. A larger cohort of patients, including a substantial number of patients receiving CYP3A4-interacting drugs, should be studied. As has been demonstrated by others, proton pump inhibitors did not influence exposure [12].

To the best of our knowledge, we conducted the first study evaluating the feasibility of capillary blood sampling of dabrafenib and trametinib as part of a clinical routine to provide the basis for generating more precise estimates for PK parameters. Our data show that most patients can perform self-sampling as 95.3% of devices were sampled correctly. However, patients who were primarily considered incapable or those who did not want to participate in the at-home sampling were not included. The low inter-occasion variability in the limited amounts of at-home samples is remarkable. Reproducibility for trametinib was poor in only one patient (DT002, Appendix A), while reproducibility for dabrafenib was excellent in the same patient. Reasons might be non-adherence or inaccuracy in the recording times of the last administration and sampling intervals. To be able to compare capillary blood concentrations to serum concentrations and to receive more precise PK parameters, we extended the existing popPK model of Balakirouchenane et al. by additionally informing it with our serum samples as well as VAMS samples collected at home and in the clinic and established a popPK model-informed VAMS-to-serum conversion model. Models for both compounds resulted in a good fit with the experimental data, indicating that the chosen conversion method from serum to blood concentrations is valid. Dabrafenib is found almost exclusively in plasma, whereas trametinib can be found at high concentrations in erythrocytes; thus, there is a need for a blood cell-to-plasma partition coefficient. The value for K_bp_ was estimated from the combined serum and VAMS dataset to be 4.62 with low IIV (CV 9.6%). Despite the low IIV, we propose to collect one paired serum and VAMS sample in the clinic and let the patient collect four additional samples at home in the same dosing interval. The paired sample can be used to estimate the individual K_bp_ rather than using the population estimate. Our model is more flexible than classical clinical validation, where only paired samples (sometimes only trough samples) are collected. This is because prior knowledge (the model) and unpaired samples (VAMS and serum concentrations) can be used to estimate the conversion model parameters. The comparison of EBE for AUC_τ_ based on the existing model of Balakirouchenane et al. and our newly developed model revealed similar results. However, in some patients, the average absolute AUC_τ_ showed relevant differences between the two models (Table 2 and Table 3). A larger number of patients and samples would be necessary to calculate a robust correlation between the two models. Additionally, some patients contributed more serum samples than VAMS samples, which might lead to bias in the comparison. However, we hypothesize that our new model is more precise because it is informed by more factors. This can only be confirmed by conducting a PK study. However, we explicitly decided to perform a real-life study as we wanted to evaluate the feasibility of micro-sampling as part of routine clinical care. Interestingly, we observed that both patients who had experienced trametinib dose reductions and received a reduced dose (DT018 and DT019, Table 3) had a considerably elevated simulated AUC_τ_ at the standard dose of 2mg q24h compared to patients receiving the standard dose. This finding indicates that this approach might be useful in identifying patients with abnormally high exposure. On the other hand, the only dabrafenib patient on a reduced daily dose (DT026, Table 2) did not show an elevated average simulated AUC_τ_ for the standard dose of 150mg q12h. It is possible that the occurrence of dose-limiting toxicity does not always result in the dose reduction of the right compound. Using a combination therapy of two compounds with partially similar side effects generally complicates the management of AE and dosage adjustments. The product information of the European Medicines Agency recommends reducing the dose for both compounds simultaneously, except for a few specific side effects that are primarily related to either dabrafenib or trametinib [40,41]. Nevertheless, our data from clinical routine show that often the dosage of only one compound is reduced. Our model can be a helpful tool to detect patients with overexposure and to decide which compound to reduce. A limitation of our study is that the assessment of AE was only conducted in parallel to serum sampling. As we hypothesize that our newly developed model is more precise than previous iterations, further research should investigate AE and dose reductions in parallel with micro-sampling to find out if this reveals different correlations. Moreover, we were not able to analyze hydroxy-dabrafenib in VAMS samples, as the methodology was not available at the time of analysis, and VAMS samples cannot be stored after extraction. The potential effects of hydroxy-dabrafenib might have been missed.

## 5. Conclusions

Our data do not support monitoring dabrafenib and trametinib in melanoma patients. However, the existing evidence for associations between toxicity and exposure in other publications and the PK data generated from our popPK model-informed VAMS-to-serum conversion model in patients receiving reduced doses of dabrafenib or trametinib suggest that monitoring may assist in making decisions regarding dose reductions due to AE. Furthermore, we were able to demonstrate that at-home sampling of capillary blood can be implemented in clinical routine and that our newly developed popPK model-informed VAMS-to-serum conversion model may be a helpful tool for receiving precise PK parameters.

## Figures and Tables

**Figure 1 cancers-14-04566-f001:**
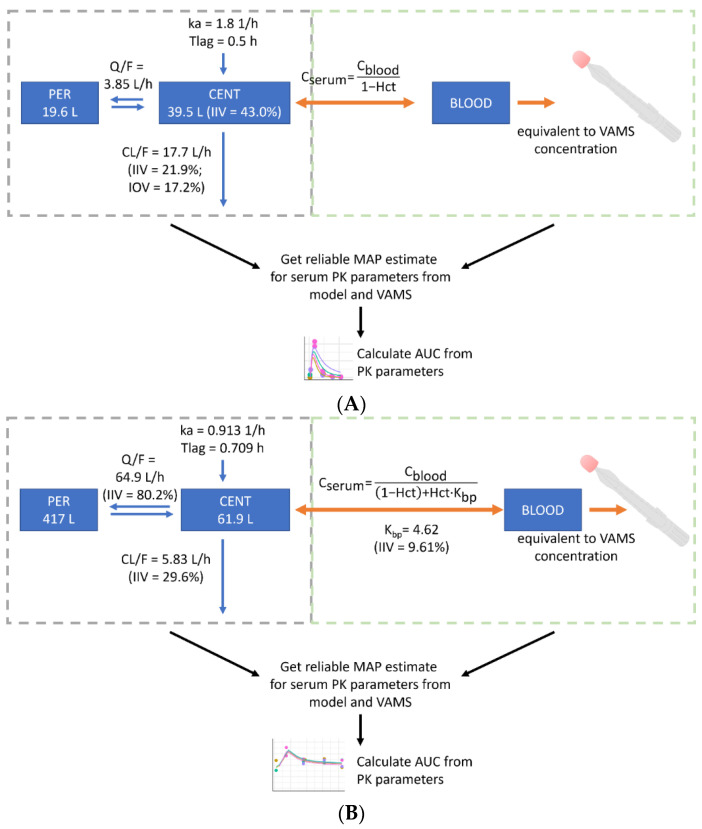
Development of the population pharmacokinetic (popPK) model-informed VAMS-to-serum conversion model. PopPK models by Balakirouchenane et al. [12] were used to generate serum maximum a posteriori (MAP) estimates from volumetric absorptive micro-sampling (VAMS) for dabrafenib (**A**) and trametinib (**B**). AUC, area under the curve; BLOOD, whole blood compartment; C_blood_, concentration in whole blood; C_plasma_, concentration in plasma; CENT, central compartment; CL/F, oral clearance from central compartment; Hct, hematocrit; IIV, inter-individual variability; IOV, inter-occasion variability; k_a_, absorption rate constant; K_bp_, partition ratio between blood cells and plasma; PER, peripheral compartment; Q/F, intercompartmental clearance; Tlag, lag time before beginning of absorption process.

**Figure 2 cancers-14-04566-f002:**
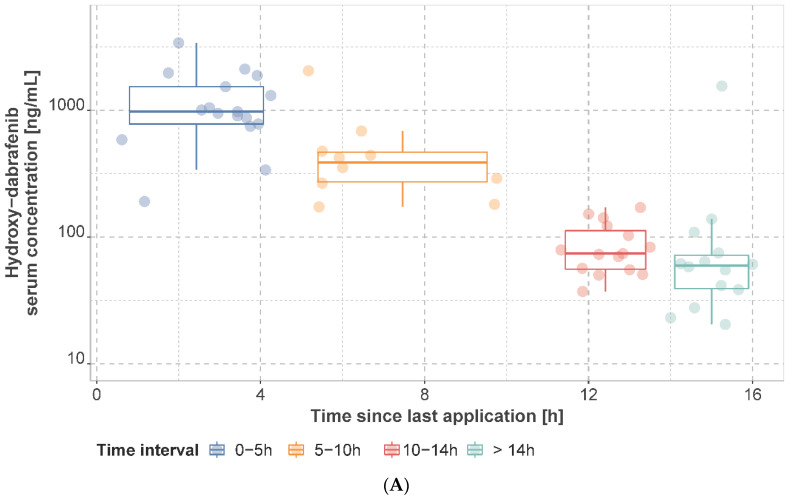
Observed hydroxy−dabrafenib (**A**) and dabrafenib (**B**) serum concentrations. Concentrations are presented as mean concentration per patient at steady state stratified by sampling time interval. Patients may have contributed samples at multiple time intervals.

**Figure 3 cancers-14-04566-f003:**
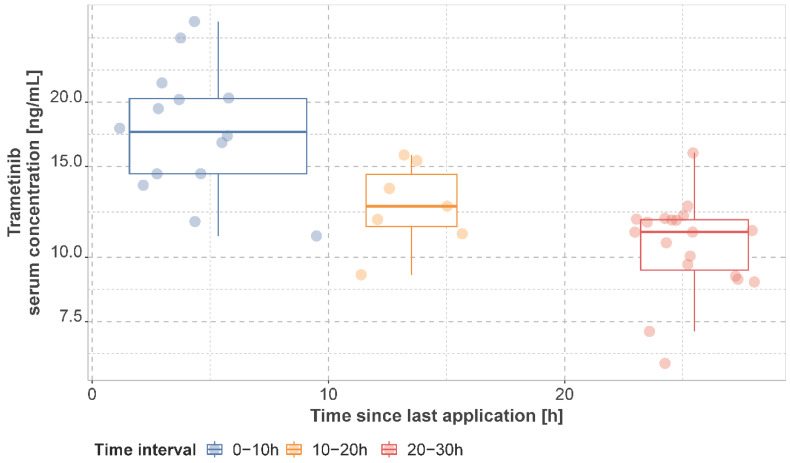
Observed trametinib serum concentrations. Concentrations are presented as mean concentration per patient in steady state stratified by sampling time interval. Patients may have contributed samples at multiple time intervals.

**Figure 4 cancers-14-04566-f004:**
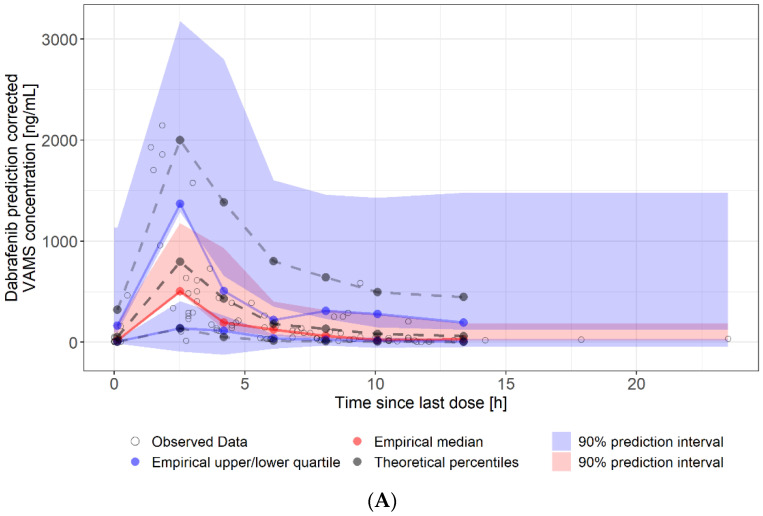
Visual predictive check of at-home sampled VAMS concentrations. (**A**): dabrafenib VAMS concentrations (90 samples, eight patients). (**B**): trametinib VAMS concentrations (84 samples, seven patients). Solid lines represent the 5th (lower blue), 50th (red), and 95th (upper blue) percentiles of the observed data. Shaded regions represent the 90% confidence intervals surrounding the 5th, 50th, and 95th percentiles from the predicted data. The plot demonstrates that the model predictions captured the majority of observed dabrafenib and trametinib concentrations within the 5th and 95th percentiles of the simulated values.

**Table 1 cancers-14-04566-t001:** Baseline patient demography.

Patient Characteristic	No. of Patients	%
**Total**	27	
**Age, median (range)**	66 (40–76)	
**Weight (kg), mean (range)**	81.2 (54.4–115)	
**Height (cm), mean (range)**	174.4 (164–186) ^a^	
**BMI (kg/m²), median (range), (IQR)**	26.7 (18.3–39.3) ^a^	
**Gender**		
Male	19	70.4
Female	8	29.6
**Ethnicity**		
Caucasian	27	100
**Smoking status**		
Smoker	4	14.8
Non-smoker	22	81.5
Unknown	1	3.7
**CYP inhibitors**	**No. of occasions (patients)**	
Strong CYP2C8 inhibitor		
1 inhibitor	265 (27)	95.3
none	13 (2)	4.7
Moderate CYP2C8 inhibitor		
2 inhibitors	5 (1)	1.8
1 inhibitor	94 (10)	33.8
none	179 (18)	64.4
Strong or moderate CYP3A4 inhibitor		
1 inhibitor	2 ^b^ (2)	0.7
none	276 (27)	99.3
**P-gp inhibitors**		
3 inhibitors	19 (3)	6.9
2 inhibitors	41 (5)	14.7
1 inhibitor	89 (9)	32.0
none	129 (15)	46.4
**P-gp inducers**		
2 inducers	6 ^c^ (1)	2.2
1 inducer	105 (11)	37.8
none	167 (18)	60.0
**Proton pump inhibitor**		
Yes	84 (9)	30.2
No	194 (21)	69.8
**AJCC stage** ^d^	**No. of patients**	
*Stage IIIB*	5	18.5
*Stage IIIC*	6	22.2
*Stage IIID*	1	3.7
*Stage IV*	15	55.6
**Duration of treatment at enrollment (days), median (range)**		
Dabrafenib	146 (11–1494)	
Trametinib	146 (11–1466)	
Median time in study (range)	324 (26–714)	

^a^ height and therefore BMI was not available for one patient; ^b^ one occasion with strong and one with moderate CYP3A4 inhibitor; ^c^ one strong and one moderate P-gp inducer; ^d^ refers to staging at initiation of dabrafenib or trametinib therapy. BMI, body mass index; IQR, interquartile range; CYP, cytochrome P450; P-gp, P-glycoprotein; AJCC, American Joint Committee on Cancer.

**Table 2 cancers-14-04566-t002:** MAP estimates for dabrafenib serum pharmacokinetic parameters using at-home sampled VAMS.

ID	Occasion	Dose [mg/12 h]	Ind V1/F [L]	Ind CL/F [L/h]	Simulated AUC_τ_ for 150 mg q12h [ng·h/L]	Average Simulated AUC_τ_ for 150 mg q12h Using at-Home VAMS [ng·h/L]	Average Simulated AUC_τ_ for 150 mg q12h Using Untimed Serum Sampling [ng·h/L]
DT002	1	150	30.8	13.5	11,125	9045	7809
DT002	2	150	30.8	17.2	8815		
DT002	3	150	30.8	21.1	7195		
DT005	1	150	24.7	20.7	7345	7345	6240
DT010	1	150	14.3	18.5	8254	8443	3610
DT010	2	150	14.3	33.3	4530		
DT010	3	150	14.3	12.1	12,544		
DT014	1	150	30.2	30.1	5005	5005	3145
DT018	1	150	82.5	24.7	6081	6259	5944
DT018	2	150	82.5	14.9	10,223		
DT018	3	150	82.5	37.5	4046		
DT018	4	150	82.5	32.3	4686		
DT019	1	150	39.5	33.1	4563	8925	5709
DT019	2	150	39.5	15.8	9501		
DT019	3	150	39.5	9.8	15,347		
DT019	4	150	39.5	24.2	6288		
DT026	1	100	40.3	33.3	4530	4035	4999
DT026	2	100	40.3	41.1	3674		
DT026	3	100	40.3	38.7	3903		
DT027	1	150	71.5	28.6	5287	5302	6350
DT027	2	150	71.5	23.2	6457		
DT027	3	150	71.5	36.5	4164		

Ind V1/F, individual volume of distribution; Ind Cl/F, individual oral clearance.

**Table 3 cancers-14-04566-t003:** MAP estimates for trametinib serum pharmacokinetic parameters using at-home sampled VAMS.

ID	Dose [mg/24 h]	Ind Q/F [L/h]	Ind CL/F [L/h]	Ind K_bp_	Simulated AUC_τ_ for 2 mg q24h Using at-Home VAMS [ng·h/L]	Simulated AUC_τ_ for 2 mg q24h Using Untimed Serum Sampling [ng·h/L]
DT002	2	97.65	6.07	4.84	326	358
DT010	2	125.55	6.63	4.98	299	300
DT014	2	129.17	8.36	5.36	239	326
DT018	1	185.04	3.52	3.89	527	252
DT019	1.5	77.82	3.79	3.91	496	336
DT026	2	106.5	6.25	4.38	317	303
DT027	2	65.04	6.95	4.41	286	304

At least one paired sample was used to calculate the MAP estimate for the individual K_bp_. Since the model did not include inter-occasion variability, estimates were not different for different occasions per patient.

## Data Availability

The data presented in this study and the R-code for the VAMS-to-serum conversion model are available upon reasonable request from the corresponding author. The data are not publicly available due to ethical restrictions.

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
