# Peer review of "Monitoring of Dabrafenib and Trametinib in Serum and Self-Sampled Capillary Blood in Patients with BRAFV600-Mutant Melanoma"

_cancers, 2022, doi:10.3390/cancers14194566_

Round 1

Reviewer 1 Report

Overall the study is interesting and has clinical relevance. However I would work on the language extensively, as there are many grammar mistakes. Plus the labeling in the figures could be more clear and self-explanatory. For example, the terms "Cmax/mid-tau/trough" are not reader-friendly. Yes they're explained in the text but the terms themselves aren't generally accepted. Unnecessary terms shall be changed. For example, hydroxy-dabrafenib is unnecessary and dabrafenib is enough.

Overall the purpose of the paper is to try to improve clinical practice, so I would cut down lots of unnecessary details in the intro and discussion as well.  The current manuscript is way too wordy. 

I have also noticed that proper citations are lacking in many places. There are many sentences that sound like statements that shall have a citation followed but without. For example between line 93-102, there weren't a single citation and most sentences are vague. 

Author Response

Response to Reviewer 1

Dear Reviewer,

Thank you very much for reviewing our manuscript with the title “Monitoring of Dabrafenib and Trametinib in Serum and Self-Sampled Capillary Blood in Patients with BRAFV600-Mutant Melanoma”. Your comments have been very helpful for improving our manuscript. We have carefully considered all issues and modified our revised manuscript accordingly. In the following we would like to explain point by point all changes that have been made.

Sincerely,

Dr. Nora Isberner

Point 1: Overall the study is interesting and has clinical relevance. However I would work on the language extensively, as there are many grammar mistakes.

Response 1: Thank you for pointing out that the manuscript needs improvement regarding the language. To address this issue, we used the English language editing service offered by MDPI and additionally had the manuscript checked by a native English-speaking scientist. Changes to the manuscript were made accordingly.

Point 2: Plus the labeling in the figures could be more clear and self-explanatory. For example, the terms "Cmax/mid-tau/trough" are not reader-friendly. Yes they're explained in the text but the terms themselves aren't generally accepted.

Response 2: We checked the figure labeling of all figures. Labeling of Figures 2 and 3, Supplementary Tables S12 and S13, and of Supplementary Figures S1, S2, S3, and S4 were modified. We made sure that the headline of each figure caption (in bold) is short and comprehendible. However, in our opinion it is necessary to provide additional information for each figure to make sure that the reader understands the figure’s content. The terms Cmax, mid-tau and trough were deleted in the manuscript and in Figures 2 and 3. Instead, the time intervals post-dose are now presented in the legend of both figures.

Point 3: Unnecessary terms shall be changed. For example, hydroxy-dabrafenib is unnecessary and dabrafenib is enough.

Respone 3: Hydroxy-dabrafenib is one of the active metabolites of dabrafenib and is a different compound. Therefore, the two terms are not used interchangeably. In our study, both compounds were analyzed and associations for both compounds with patient covariates and adverse events were investigated independently.

Point 4: Overall the purpose of the paper is to try to improve clinical practice, so I would cut down lots of unnecessary details in the intro and discussion as well.  The current manuscript is way too wordy. 

Reviewer 2 Report

The submitted manuscript is a very interesting example of the article taking into consideration the exposure and dabrafenib on toxicity and patient characteristics. The Introduction is well organized and properly described. 

Material and methods are well and adequately described. The statistical analysis is well performed and correct tests were used to evaluate the significance of the results. However, the Reviewer is concerned about the logistic regression analysis. The Reviewer would like to ask if the Authors used validation of the model by dividing the cases into tested and validated group? If yes, please note this in the manuscript and mention the ratio of the number of patients in tested and validated group. Additionally, please note the AUC value for logistic regression model for tested group and value of Hosmer-Lemeshov test.  Summarizing, the Reviewer believes that the number of patients was too low to use logistic regression for the analyses and thus p-value is higher than 0.09 - maybe enrolling more patients into this trial and use Propensity Score Matching for statistical analysis would improve the results.

The Results are properly described. 

The Discussion section is prepared in a thoughtful way and a sufficient number of articles was cited.

Summarizing, I recommend this manuscript for publication when the Authors comment the abovementioned issues.

Author Response

Response to Reviewer 2

Dear Reviewer,

Thank you very much for reviewing our manuscript with the title “Monitoring of Dabrafenib and Trametinib in Serum and Self-Sampled Capillary Blood in Patients with BRAFV600-Mutant Melanoma”. Your comments have been very helpful for improving our manuscript. We have carefully considered all issues and modified our revised manuscript accordingly. In the following we would like to explain point by point all changes that have been made.

Sincerely,

Dr. Nora Isberner

Reviewer’s comment: The submitted manuscript is a very interesting example of the article taking into consideration the exposure and dabrafenib on toxicity and patient characteristics. The Introduction is well organized and properly described.  Material and methods are well and adequately described. The statistical analysis is well performed and correct tests were used to evaluate the significance of the results.

However, the Reviewer is concerned about the logistic regression analysis. The Reviewer would like to ask if the Authors used validation of the model by dividing the cases into tested and validated group? If yes, please note this in the manuscript and mention the ratio of the number of patients in tested and validated group. Additionally, please note the AUC value for logistic regression model for tested group and value of Hosmer-Lemeshov test.  Summarizing, the Reviewer believes that the number of patients was too low to use logistic regression for the analyses and thus p-value is higher than 0.09 - maybe enrolling more patients into this trial and use Propensity Score Matching for statistical analysis would improve the results.

The Results are properly described. The Discussion section is prepared in a thoughtful way and a sufficient number of articles was cited. Summarizing, I recommend this manuscript for publication when the Authors comment the abovementioned issues.

Response: Thank you very much for your detailed comments about the statistical analyses used in our study. Our study is exploratory since it includes a limited number of patients (n = 27). Since it is a single-center study, it was not possible to include more patients during the study period. A substantial number of people are diagnosed with malignant melanoma in Germany every year. However, most patients are diagnosed at early stages and do not require treatment with dabrafenib and trametinib. Moreover, not all patients with advanced or metastatic melanoma receive kinase inhibitors. Depending on tumor stage, potential to operate the tumor and metastases, and other tumor and patient characteristics, surgery, immunotherapy and chemotherapy are alternatives to treatment with dabrafenib and trametinib. Unfortunately, it is not possible to enroll more patients since the trial is already closed and necessary structures to enroll more patients no longer exist (e.g., approval of the ethics commission, staff for collection and analysis of samples, instructing patients, data management, etc.). Moreover, our cohort is rather heterogenous, for example regarding tumor stage or duration of treatment at study enrollment. Therefore, validation of the model by dividing cases into two groups (tested vs. validated) or propensity score matching was not performed. As recommended, we additionally performed the Hosmer-Lemeshow test. Results showed that our logistic regression model is suitable. Changes were made in the introduction (lines 198-200 tracked-changes version) and results (lines 372-375 tracked-changes version) section. Nevertheless, it should be pointed out in the manuscript that results have to be interpreted carefully due to the low number of patients. This was added in the discussion (lines 472-473 tracked-changes version and lines 495-497 tracked-change version). We are confident that our data and our newly developed model could be the basis for future multi-center investigations including more patients and that the proposed statistical tests could then be applied.

Reviewer 3 Report

The authors have done nice work by monitoring of Dabrafenib and Trametinib in serum and capillary blood in melanoma patients.

This study gives very useful information and may add to the literature including new understanding of the phenomena in the content area.

However, the final edit and language check of the manuscript is needed before accepted for the publishing.

And do authors feel that such study will also highlight the overall efficacy and safety of the drug(s).

Author Response

Response to Reviewer 3

Dear Reviewer,

thank you very much for reviewing our manuscript with the title “Monitoring of Dabrafenib and Trametinib in Serum and Self-Sampled Capillary Blood in Patients with BRAFV600-Mutant Melanoma”. Your comments have been very helpful for improving our manuscript. We have carefully considered all issues and modified our revised manuscript accordingly. In the following we would like to explain point by point all changes that have been made.

Sincerely,

Dr. Nora Isberner

Point 1: The authors have done nice work by monitoring of Dabrafenib and Trametinib in serum and capillary blood in melanoma patients. This study gives very useful information and may add to the literature including new understanding of the phenomena in the content area. However, the final edit and language check of the manuscript is needed before accepted for the publishing.

Response 1: Thank you for pointing out that the manuscript needs improvement regarding the language. To address this issue, we used the English language editing service offered by MDPI and additionally had the manuscript checked by a native English-speaking scientist. Changes to the manuscript were made accordingly.

Point 2: And do authors feel that such study will also highlight the overall efficacy and safety of the drug(s).

Response 2: Our study design does not allow to draw conclusions about the overall efficacy of dabrafenib and trametinib as we did not analyze outcome. Results on the safety of both compounds in our cohort (treatment discontinuations and dosage adjustments in our cohort compared to the phase III trial cohort) are mentioned in the discussion (lines 466-472 tracked-changes version). The data suggest that dose reductions and treatment continuations in our cohort occur as frequently as in the phase III trial. However, the aim of our study was not to generally assess safety. For this purpose, a different study design should have been chosen including only patients at initiation of therapy and assessing all adverse events occurring during the study period (including adverse events not requiring visits to the clinic and including adverse events requiring hospital admission). Therefore, results on safety in our cohort were not explicitly highlighted in our manuscript.

Round 2

Reviewer 1 Report

Sufficient for publication